# Research on an Improved Rule-Based Energy Management Strategy Enlightened by the DP Optimization Results

Dapai Shi [1], Junjie Guo [1], Kangjie Liu [2,*], Qingling Cai [1], Zhenghong Wang [1] and Xudong Qu [1]

1    Hubei Key Laboratory of Power System Design and Test for Electrical Vehicle, Hubei University of Arts and Sciences, Xiangyang 441053, China; shidapai@hbuas.edu.cn (D.S.); 202208551048@hbuas.edu.cn (J.G.); 202208551040@hbuas.edu.cn (Q.C.); 202208551067@hbuas.edu.cn (Z.W.); 202208551062@hbuas.edu.cn (X.Q.)
2    School of Automotive Studies, Tongji University, Shanghai 201804, China
*    Correspondence: liukj21@tongji.edu.cn

**Abstract:** Plug-in hybrid electric vehicles (PHEVs) have gradually become an important member of new energy vehicles because of the advantages of both electric and hybrid electric vehicles. A fast and effective energy management strategy can significantly improve the fuel-saving performance of vehicles. By observing the dynamic programming (DP) simulation results, it was found that the vehicle is in the charge-depleting mode, the state of charge (SOC) drops to the minimum at the end of the journey, and the SOC decreases linearly with the mileage. As such, this study proposed an improved rule-based (IRB) strategy enlightened by the DP strategy, which is different from previous rule-based (RB) strategies. Introducing the reference SOC curve and SOC adaptive adjustment, the IRB strategy ensures that the SOC decreases linearly with the driving distance, and the SOC drops to the minimum at the end of the journal, similar to the result of the DP strategy. The fuel economy of PHEV in the RB and DP energy management strategies can be considered as their worst-case and best-case scenarios, respectively. The simulation results show that the fuel consumption of the IRB strategy under the China Light-duty Vehicle Test Cycle is 3.16 L/100 km, which is 7.87% less than that of the RB strategy (3.43 L/100 km), and has reached 44.41% of the fuel-saving effect of the DP strategy (2.84 L/100 km).

**Keywords:** plug-in hybrid electric vehicle; energy management strategy; dynamic programming strategy; rule-based strategy; the reference SOC





## 1. Introduction

Energy shortage and environmental pollution have promoted the rapid development of new energy vehicles. However, due to the limitations of battery technology and charging infrastructure, electric vehicles (EVs) are difficult to popularize at a large scale in a short time. In view of this, plug-in hybrid electric vehicles (PHEVs), which have both EV and hybrid electric vehicle advantages, have naturally become an ideal model for the transition from internal combustion engine vehicles to EVs. PHEVs typically consist of series [1], parallel [2], series–parallel [3], and power-split configurations [4]. Among parallel configurations, the P2 configuration PHEV has become one of the leading members because of its simple structure and strong power [5]. The P2 configuration PHEV has a mechanical connection between the engine and the wheels, which makes the fuel economy of the vehicle greatly affected by the driving conditions.

For this reason, the energy management strategy based on driving condition identification is proposed in [6–8], and a good fuel economy is achieved. However, when the driving condition changes dramatically, the identification result of the driving condition will lag behind the actual driving condition to a certain extent [9–11]. The energy management strategy based on condition prediction is proposed in [12–15], which usually can only accurately predict the speed in the next few seconds (<10 s). Therefore, it can be regarded

as an instantaneous optimization management strategy, so fuel economy improvement is limited [16,17]. The energy management strategy based on an intelligent transportation system is proposed in [18,19]. The overall planning of power battery power has achieved good fuel economy, but it depends on the current construction of an intelligent transportation system. The energy management strategy based on intelligent network connection is proposed in [20–22], which is similar to an intelligent transportation system, and can also ensure that vehicles achieve good economy. Considering the limitations of the above energy management strategy, it is necessary to design a fast, efficient and stable energy management strategy.

The RB strategy based on the optimization strategy is proposed in [23–26]. Du et al. optimized the logic threshold value of the RB strategy by using the improved NSGA_II algorithm to obtain the optimal logic threshold value, and combined the optimized RB strategy with the minimum equivalent fuel consumption strategy, and a real-time control strategy for a multi-mode hybrid electric vehicle is proposed [23]. Li et al. improved the adaptability of the RB strategy by optimizing parameters under multiple historical driving cycles and obtaining a new RB strategy [25]. Two rule-based control strategies, i.e., the Engine-Dominant and Motor-Dominant strategies, are proposed for a power-split configuration and compared regarding fuel consumption and emissions under a city-highway combined driving cycle [26]. Using the off-line optimization results, the new RB can improve the vehicle's fuel economy, but there may be a problem of driving condition adaptability [27–29]. RB is not fuel efficient but requires less computing power, and DP is fuel efficient, but computation might be more complex. Hence, an intermediary strategy is required to combine the advantages of both. As such, this study proposed an IRB strategy enlightened by the DP strategy.

In summary, this study proposes an IRB energy management strategy based on the enlightenment of DP optimization results. By observing the simulation results of the DP strategy and the RB strategy, it is found that the DP strategy is in the charge-depleting (CD) mode throughout the journey, and at the end of the journey, the SOC of the power battery drops to the minimum, and the SOC is inversely linear with time and driving distance. Considering that the actual driving time is often limited by traffic conditions, taking the driving time as a reference may lead to a significant error. The linear programming of the power battery SOC is carried out by using the driving distance, and the IRB strategy is established. By introducing the reference SOC curve and SOC adaptive adjustment, the IRB strategy ensures that the SOC decreases linearly with the driving distance, and the SOC drops to the minimum at the end of the journal, similar to the result of the DP strategy.

The rest of this article is arranged as follows. Section 2 introduces the process of establishing the simulation model; the energy management strategy is given in Section 3; the last part is the summary and conclusion.

## 2. The PHEV Model of the RB Energy Management Strategy

### 2.1. The Vehicle Model

The research object of this study is the P2 configuration PHEV, whose engine is coaxially connected with the start–stop-integrated motor (ISG) and can drive the vehicle alone or together. The power system structure and vehicle parameters are shown in Figure 1 and Table 1, respectively. There is a mechanical connection between the engine and the wheels of the P2 configuration PHEV, so the vehicle's fuel economy is greatly affected by the driving conditions.

**Table 1.** Vehicle parameters of PHEV with P2 configuration.

| Components | Parameter | Value |
|---|---|---|
| Vehicle | Mass | 1449 kg |
| | Wheelbase | 2.600 m |
| | Front area | 2.25 m$^2$ |

**Table 1.** *Cont.*

| Components | Parameter | Value |
|---|---|---|
| Engine | Displacement<br>Maximum speed<br>Maximum torque | 1.0 L<br>6000 rpm<br>170 N m at 4000 rpm |
| Motor | Maximum torque<br>Rated voltage | 140 N m<br>288 V |
| Power battery | Capacity<br>Voltage | 35 Ah<br>300 V |
| Gearbox | Speed ratio | 3.84/2.43/1.71/1.27/1/0.82/0.69 |
| Main reducer | Speed ratio | 3.94 |

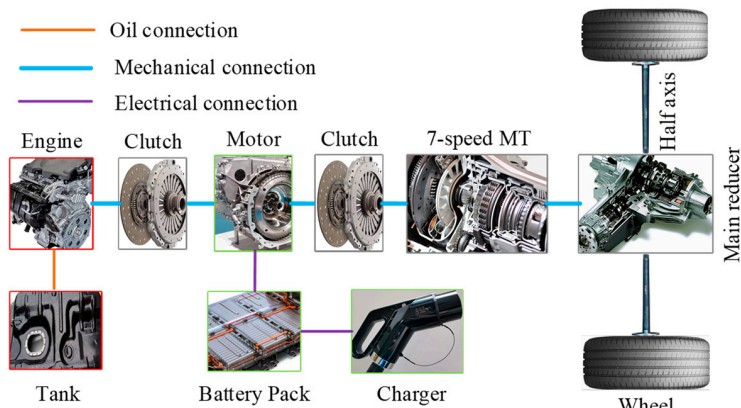

**Figure 1.** Structure of an PHEV dynamic system with P2 configuration.

### 2.2. The Engine Fuel Consumption Model

The engine fuel consumption in the engine static diagram model can be obtained by looking up the table of engine speed and torque, which has the characteristics of fast calculation and high accuracy [30]. The instantaneous fuel consumption of the engine can be calculated by the following formula:

$$\dot{m}_f = f(T_e, n_e) \tag{1}$$

In the formula, $\dot{m}_f$ is the instantaneous fuel consumption of the engine, $T_e$ and $n_e$ are the torque and speed of the engine, respectively, and the corresponding relationship between them can be obtained by looking up the table of the universal characteristic curve of the engine, which is shown in Figure 2.

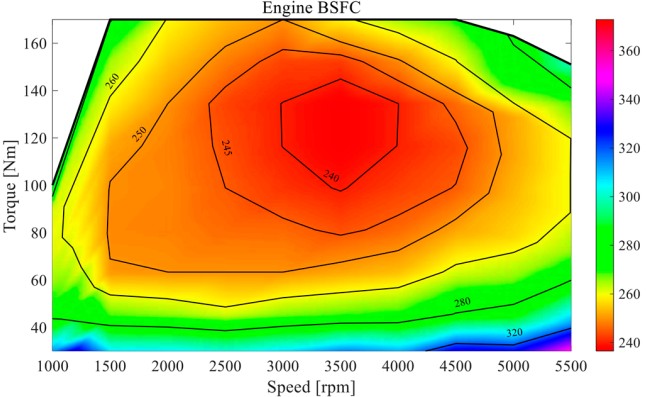

**Figure 2.** The universal characteristic curve of the engine.

### 2.3. The Battery Model

Usually, the purpose of establishing the power battery model is to predict the change in the $SOC$ of the power battery under a given electric load [31].

When using the ampere-hour method, the power battery $SOC$ can be expressed as:

$$SOC = SOC_i - \frac{1}{3600Q_b} \int I_b dt \tag{2}$$

Among them, $SOC_i$ is the initial $SOC$ and is dimensionless, $Q_b$ is the power battery capacity with Ah, and $I_b$ is the power battery current.

The characteristics of open circuit voltage and equivalent internal resistance of power battery varying with $SOC$ at an ambient temperature of 25 °C are shown in Figure 3. When the $SOC$ is less than 0.6, there will be a rapid increase in the internal resistance of the power battery. According to Joule's law, an increase in internal resistance will increase the energy loss of the power battery.

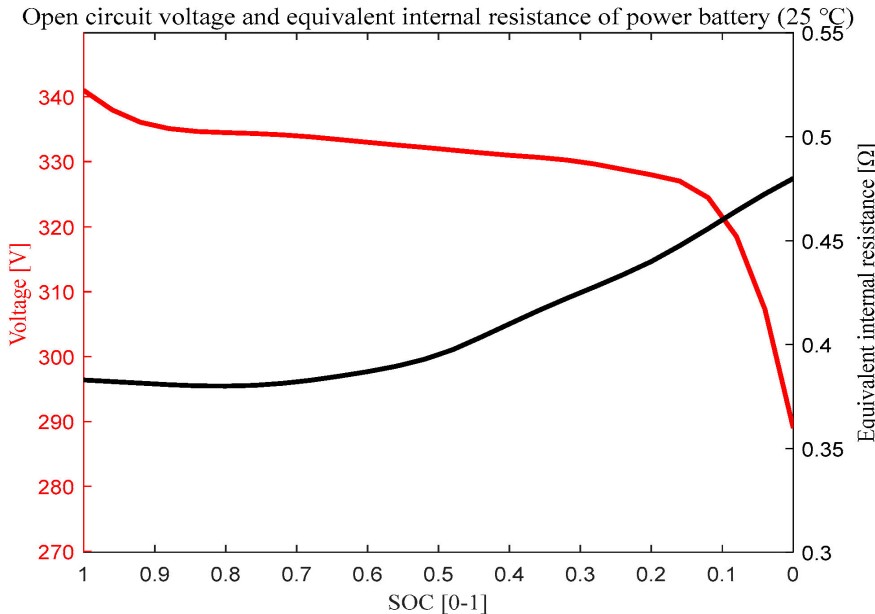

**Figure 3.** Relation curve between open circuit voltage, equivalent internal resistance and $SOC$.

### 2.4. The ISG Model

According to the working mode of $ISG$, the power of the motor can be calculated by the following formula:

$$P_{isg} = \begin{cases} \frac{n_{isg} T_{isg}}{9550 \eta_{isg}}, & T_{isg} \geq 0 \\ \frac{n_{isg} T_{isg}}{9550} \eta_{isg}, & T_{isg} < 0 \end{cases} \tag{3}$$

Among them, $P_{isg}$ is the motor power; $n_{isg}$ is the motor speed; $T_{isg}$ is the motor torque, which defines that the torque is positive when ISG is driven and negative when ISG is generated; $\eta_{isg}$ is the motor efficiency under the corresponding speed and torque.

$\eta_{isg}$ can be calculated by the following formula

$$\eta_{isg} = f\left(T_{isg}, n_{isg}\right) \tag{4}$$

In the formula, $\eta_{isg}$ is the ISG efficiency and $T_{isg}$ and $\eta_{isg}$ are the torque and speed of the engine, respectively, as shown in Figure 4.

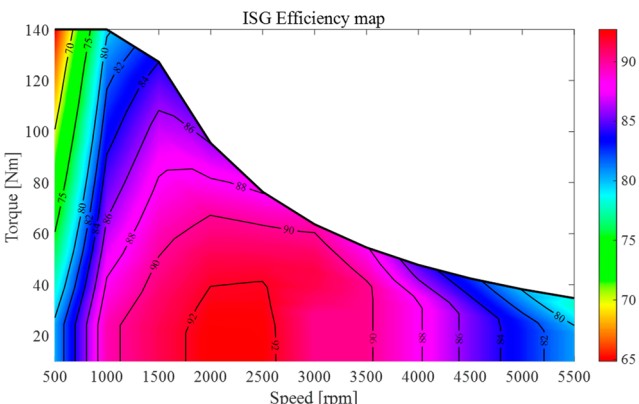

**Figure 4.** ISG map with 93% efficiency.

### 3. Energy Management Strategy

#### 3.1. The RB Strategy

Under the RB strategy, PHEV usually has two working modes: the CD mode and the Charge Sustaining (CS) mode. In the CD mode, the vehicle is mainly driven by ISG, and the engine is driven only when the ISG cannot meet the power demand; in the CS mode, the vehicle is mainly driven by the engine, and the ISG only participates in the work when the engine cannot meet the power demand or when the braking energy is recovered. The designed RB energy management strategy is mainly based on the SOC switching mode. When SOC is greater than 0.3, the vehicle is in the CD mode; when SOC is less than 0.3, the vehicle enters the CS mode to ensure that SOC can be maintained near 0.3. Choosing SOC equal to 0.3 as the switching point of the two operating modes is mainly due to the effect of equivalent internal resistance of the power battery. When the SOC is lower than 0.3, the equivalent internal resistance of the power battery will increase rapidly.

#### 3.2. The DP Strategy

The required torque at the input of the gearbox can be calculated reversely from the tangential force of the wheel. According to the given driving conditions, based on the inverse dynamics model, the required torque $T_{req}$ of the vehicle can be calculated as

$$T_{req} = F_t \cdot r \cdot i_0 \cdot \frac{i_g}{\eta_t} \qquad (5)$$

Among them, $F_t$ is the tangential driving force generated by the driving wheel, $r$ is the wheel radius, $i_0$ is the speed ratio of the main reducer, $i_g$ is the speed ratio of the transmission, and $\eta_t$ is the transmission efficiency of the mechanical system.

According to the longitudinal dynamic equation, $F_t$ can be expressed as

$$F_t = mg\,f cos\alpha_{slap} + C_D A \frac{\rho}{2} u^2 + mgsin\alpha_{slap} + \sigma m \frac{d}{dt} v \qquad (6)$$

where $m$ is the vehicle mass, $g$ is the gravity acceleration, $f$ is the rolling resistance coefficient, $\alpha_{slap}$ is the road slope, $C_D$ is the air resistance coefficient, $A$ is the windward area, $\rho$ is the air density, $u$ is the relative speed, $\sigma$ is the rotation mass conversion coefficient, and $v$ is the speed.

In addition, the required torque is jointly provided by the engine and ISG, and can also be expressed as

$$T_{req} = T_e + T_{isg} \qquad (7)$$

The DP strategy solves for each instantaneous engine torque $T_e$ and ISG torque $T_{isg}$ for the global driving conditions to ensure that the global fuel consumption is minimized in the feasible domain, so the optimization objective can be expressed as

$$minJ\left(t_f\right) = \int_{t_0}^{t_f} \dot{m}_f(u(t))dt \tag{8}$$

where $u(t)$ is the control quantity, and the engine torque is selected.

In the DP strategy, SOC is generally taken as the state variable $x$, so the state equation of the whole process can be described as

$$\dot{x} = \dot{SOC} = -\frac{I_b}{Q_b} = -\frac{V_{oc} - \sqrt{V_{oc}^2 - 4P_b R_0}}{2R_0 Q_b} \tag{9}$$

Among them, $V_{oc}$, $R_0$ and $P_b$ are the open circuit voltage, the internal resistance and the output power of the power battery, respectively.

The constraints of a DP strategy are usually physical constraints of key components, which can be expressed as

$$\begin{cases} SOC \in [SOC_{min}, SOC_{max}] \\ T_e = [T_{emin}, T_{emax}] \\ n_{emin} < n_e < n_{emax} \\ T_{isg} = \left[T_{isg\,min}, T_{isg\,max}\right] \\ n_{isg\,min} < n_{isg} < n_{isg\,max} \end{cases} \tag{10}$$

where $SOC_{min}$ and $SOC_{max}$ are the minimum and maximum usage limits of SOC, respectively; $T_{emin}$ and $T_{emax}$ are the minimum and maximum torque of the engine, respectively; $n_{emin}$ and $n_{emax}$ are the minimum and maximum speed of the engine, respectively; $T_{isg\,min}$ and $T_{isg\,max}$ are the minimum and maximum torque of the ISG, respectively; $n_{isg\,min}$ and $n_{isg\,max}$ are the minimum and maximum speed of the ISG, respectively.

### 3.3. Simulation Results

The script of the RB and DP strategies is written in MATLAB software. The simulation condition is the China Light-duty Vehicle Test Cycle (CLTC). The period of CLTC is the 1800s, and its length is 14.48 km. Since the mileage of the electric driving of the PHEV is usually larger than that of the 50 km, 10 fold the mileage of the CLTC is carried out. Considering the equivalent internal resistance of the power battery, the values of $SOC_{max}$ and $SOC_{min}$ min are 0.7 and 0.3, respectively. The starting and ending SOC of RB and DP energy management strategy are 0.7 and 0.3, respectively.

Figure 5a shows the SOC decline of RB and DP strategies under 10*CLTC driving conditions. The simulation results are also listed in Table 2. Under the 10*CLTC condition, the SOC of the DP strategy is linear with time, and the SOC drops to the minimum at the end of the journey. On the other hand, when the RB strategy is in the CD mode, SOC decreases linearly with time, but when entering the CS mode, SOC fluctuates around the minimum value of 0.3. In addition, in terms of fuel consumption, the DP strategy has reduced 17.72% compared with RB, and the corresponding power efficiency has increased by 17.15%. In addition, it is known from the equivalent internal resistance characteristics of the power battery: the equivalent internal resistance of power battery near the minimum value of SOC is larger, and the corresponding energy loss is also larger, which may be one of the reasons for the more fuel consumption of RB. Therefore, the optimal fuel economy of the DP strategy is highly related to the overall planning of power battery power.

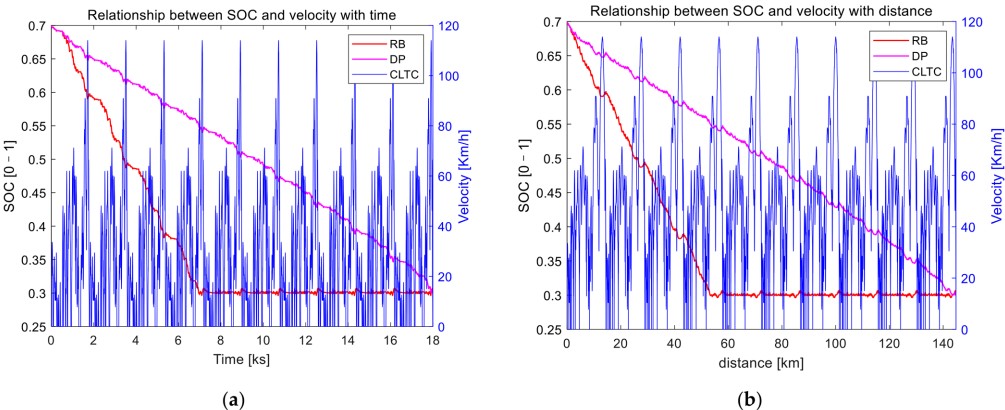

**Figure 5.** (**a**) The relationship between SOC and time. (**b**) The relationship between SOC and distance.

**Table 2.** Simulation settings and results of RB and DP strategies under CLTC.

| Control Strategy | RB | DP |
| --- | --- | --- |
| Initial SOC (%) | 70 | 70 |
| Terminal SOC (%) | 30.72 | 30.01 |
| Fuel consumption (L/100 km) | 3.43 | 2.84 |
| Electricity consumption (kW h/100 km) | 2.74 | 3.21 |
| Electricity utilization rate (%) | 0 | 17.15 |
| Fuel-saving rate (%) | 0 | 17.72 |

Figure 5b shows the relationship between SOC and distance under the DP strategy. The DP strategy shows a linear relationship between SOC and driving distance. When the RB strategy is still in the CD mode, SOC decreases linearly with driving distance, but when entering the CS mode, SOC fluctuates around the minimum value of 0.3 with distance.

However, under the actual driving conditions, the driving time is complex and changeable relative to driving distance. The following is mainly according to the inspiration of the DP strategy that SOC decreases linearly with the driving distance to improve the RB strategy.

## 4. The IRB Strategy

After the driving conditions are known, the DP strategy will search in the feasible region for the state space (SOC planning) that minimizes global fuel consumption. Although it is mainly collected in the time domain, it also shows a certain law in the space domain (SOC decreasing linearly with driving distance) and reaches the minimum value at the end of the journey. The full use of the electric energy stored in the battery package improves the fuel economy of the PHEV. Inspired by this, the linear SOC reference curve be added to the original RB strategy to obtain the IRB strategy, which can adaptively adjust the SOC of the power battery. The IRB strategy mainly comprises power battery SOC linear programming and adaptive adjustment. They will be introduced in detail as follows.

### 4.1. Power Battery Linear Programming

With the development of intelligent transportation technology and big data, the driving distance of a single trip and the whole day has been mastered in advance [32]. For this reason, the use of power battery can be linearly planned according to driving distance to improve the vehicle's fuel economy.

The design method of reference SOC is given as follows.

$$SOC_r = SOC_0 - \frac{SOC_0 - SOC_{min}}{D_t} \cdot D_i \tag{11}$$

where $SOC_r$ and $SOC_0$ represent the referenced and initial SOC, respectively. $D_t$ and $D_i$ are the total distance traveled and current distance traveled, respectively.

### 4.2. Adaptive Adjustment

To ensure that the power battery SOC can drop according to the reference SOC trajectory during the actual driving, it is necessary to set a penalty function to restrain the fluctuation of the power battery SOC. Most of the common adaptive methods use PI or PID control, and although the SOC follow results are better, it will bring the problem of frequent engine start and stop. Therefore, the method of interval control is adopted. When the SOC approaches the lower limit of the reference SOC, the engine participates in the work and charges the power battery to prevent the SOC from falling too fast; when the SOC approaches the upper limit of the reference SOC, the ISG participates in the work and discharges the power battery so that the SOC decreases rapidly.

The adaptive adjustment method can be written as

$$M_{mode} = \begin{cases} Electric, \ SOC > SOC_{ru} \\ Hybird, \ SOC < SOC_{rl} \\ Hold \ on, SOC_{rl} < SOC < SOC_{ru} \end{cases} \tag{12}$$

where $M_{mode}$ is the vehicle working mode, and $SOC_{ru}$ and $SOC_{rl}$ represent the upper and lower limits of the referenced SOC, respectively. Hold on mode means maintaining the previous working mode.

$SOC_{ru}$ and $SOC_{rl}$ can be written as

$$\begin{cases} SOC_{ru} = SOC_r + \Delta SOC \\ SOC_{rl} = SOC_r - \Delta SOC \end{cases} \tag{13}$$

where $\Delta SOC$ is the specific upper and lower boundary offset reference SOC.

The specific upper and lower boundary offset reference SOC depends on the power battery capacity of the vehicle. $\Delta SOC$, the upper and lower bound offset of the reference SOC curve, selected in this paper is $\pm 2\%$. The process of the IRB strategy can be described in Figure 6.

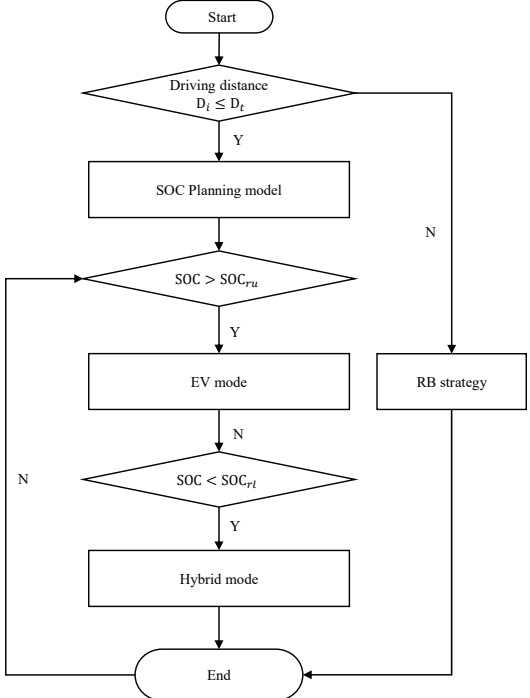

**Figure 6.** The flowchart of the IRB strategy.

### 4.3. Simulation Comparison and Analysis

The IRB simulation initial settings are the same as RB and DP. The simulation results of the three strategies are shown in the Table 3. It is worth noting that the IRB strategy shows an extraordinary fuel consumption performance, which saves 7.87% of the fuel compared with the RB strategy, which has reached 44.41% of the fuel-saving effect of the DP strategy. However, the DP strategy needs to obtain the global driving conditions, while the IRB strategy only needs to predefine the driving distance.

**Table 3.** Simulation settings and results of three strategies under CLTC.

| Control Strategy | RB | IRB | DP |
|---|---|---|---|
| Initial SOC (%) | 70 | 70 | 70 |
| Terminal SOC (%) | 30.72 | 30.94 | 30.01 |
| Fuel consumption (L/100 km) | 3.43 | 3.16 | 2.84 |
| Electricity consumption (kWh/100 km) | 2.74 | 2.83 | 3.21 |
| Electricity utilization rate (%) | 0 | 3.28 | 17.15 |
| Fuel-saving rate (%) | 0 | 7.87 | 17.72 |

For the three strategies under the CLTC condition, the SOC decrease with driving distance is shown in Figure 7. Although SOC can decrease linearly with mileage in the early stage of RB, most of the latter half of the journey is in the CS mode, and SOC fluctuates around the minimum value of 0.3 with distance.

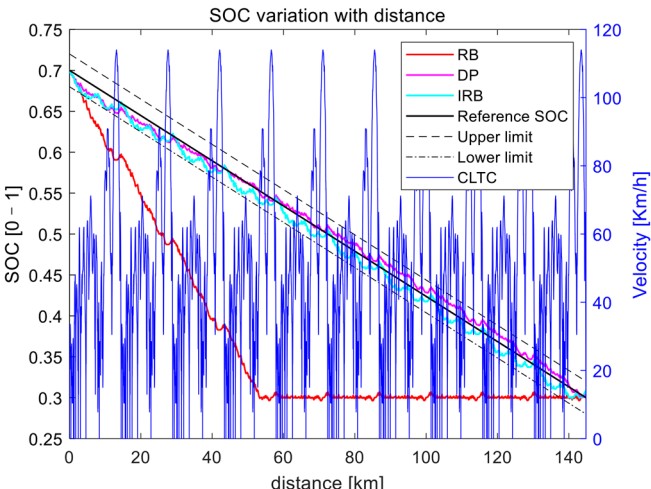

**Figure 7.** The SOC relationship of the three strategies with distance.

The engine operating points of the three strategies are shown in Figure 8. A large number of operating points of the engine under the RB strategy are distributed in the high load area, which may be one of the reasons for the more fuel consumption of the RB strategy. Compared with the RB strategy, the engine operating points under the IRB strategy are relatively uniform, and the operating points in the high-load area are reduced. Under the DP strategy, the working points of the engine are mainly concentrated in the high-efficiency area, and there are almost no working points distributed in the low-load and high load areas, which is also the essential reason for the high fuel economy of the DP strategy.

The ISG operating points of the three strategies are shown in Figure 9. The working points of the ISG under the RB strategy are relatively scattered. Compared with the RB strategy, under the IRB strategy, the ISG operating points are relatively concentrated, and the number of operating points in the high-efficiency area also increases significantly. Under the DP strategy, the ISG operating points are mainly concentrated in the high-efficiency area, and only a minimal number of operating points are distributed outside the high-efficiency area, contributing to the DP strategy's higher fuel economy.

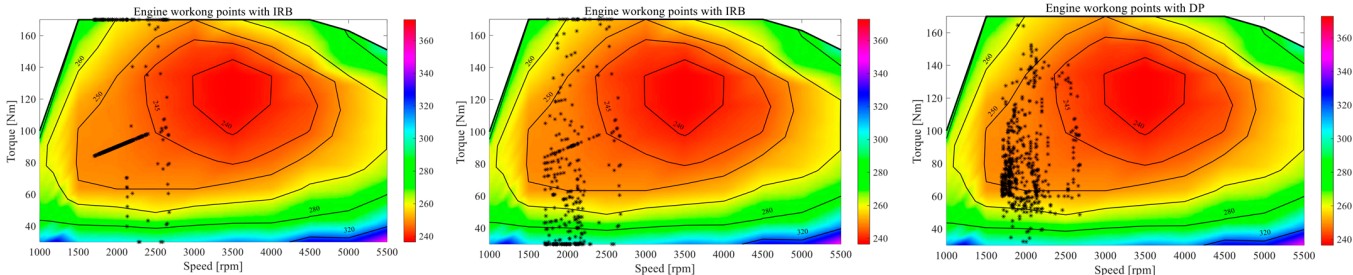

**Figure 8.** Three strategies for engine operating point distribution.

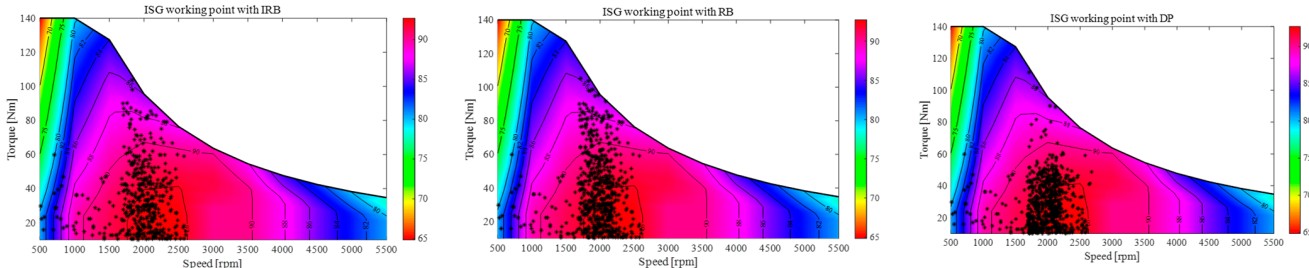

**Figure 9.** ISG operating points with three strategies.

The key component energy losses for the three strategies are shown in Figure 10. Their energy loss can be obtained by reverse solving the efficiency MAP. The instantaneous fuel consumption MAP of the engine can also be converted into efficiency MAP, which is not the focus of this research. The engine energy loss is the largest under the RB strategy, reaching 100.91 MJ, while the engine energy loss under the IRB and DP strategies is only 94.34 MJ and 83.19 MJ, respectively, which is the essential reason for the higher fuel economy of the IRB and DP strategies. Secondly, the ISG energy loss under the RB strategy is 8.30 MJ, while the ISG energy loss under the IRB and DP strategies is only 6.26 MJ and 5.75 MJ, respectively. Furthermore, the battery energy loss is 6.65 MJ under the RB strategy, while the electric energy loss is only 4.59 MJ and 4.11 MJ under the IRB and DP strategies, respectively. It is worth noting that the energy loss of the gearbox is the same under the three strategies, mainly because the shifting logic adopted by the three strategies is the same. It means the energy flowing into and out of the gearbox at each instant is the same.

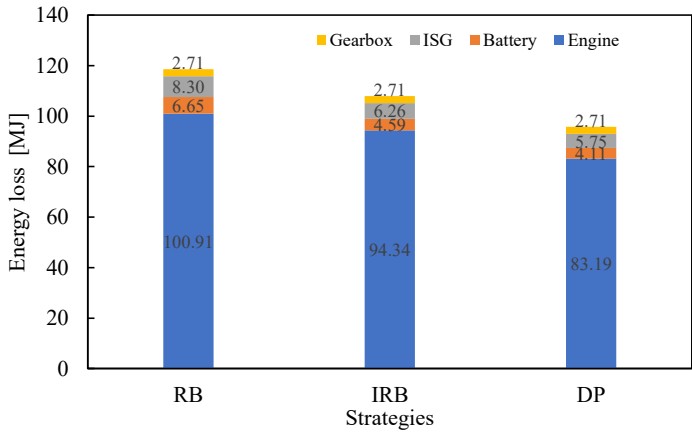

**Figure 10.** Key component's energy losses of three strategies.

Compared with the RB strategy, the IRB and DP strategies have higher fuel economy, mainly due to improved engine and ISG operating points, significantly reducing energy loss. This also avoids the rapid charging and discharge of the power battery and reduces energy loss.

## 5. Conclusions

In this study, an IRB energy management strategy based on the enlightenment of dynamic programming optimization results is proposed. By observing the economy simulation results of the DP strategy and the RB strategy, it can be found that it is in the CD model throughout the journey. At the end of the journey, SOC drops to the minimum value, and SOC is inversely linear with time and the driving distance. Therefore, this paper uses the driving distance to carry the linear programming to the power battery SOC, establishes the IRB strategy, and conducts the simulation analysis.

(1) The RB strategy prioritizes the use of power battery electricity, resulting in the vehicle always being in the CS mode at the end of the trip, increasing the secondary energy conversion loss of the vehicle. However, the DP strategy is in the CD model throughout the journey, and the SOC drops to the minimum at the end, which significantly taps the fuel-saving potential of the PHEV vehicle. However, it is necessary to predefine the global driving conditions.

(2) Similar to the DP strategy, the IRB strategy can linearly plan the power battery use so that the PHEV is in the CD mode throughout the journey, reducing the vehicle's secondary energy conversion loss and improving the vehicle's fuel economy. It only needs to obtain the driving distance.

(3) The fuel consumption of the IRB strategy under CLTC condition is 3.16 L/100 km, which is 7.87% less than that of the RB strategy (3.43 L/100 km). It has reached 44.41% of the fuel-saving effect of the DP strategy (2.84 L/100 km).

(4) Compared with the RB strategy, the IRB and DP strategies have higher fuel economy, mainly due to the improvement of the engine and ISG operating point, which not only significantly reduces the energy loss but also avoids the rapid charging and discharge of power battery, and further reduces the energy loss.

In view of the effectiveness of the IRB strategy, the next step of the research will consider more road condition information, such as slope and average speed, to further improve the fuel-saving potential of this strategy.

**Author Contributions:** Conceptualization, D.S., K.L. and Q.C.; methodology, J.G., Z.W. and X.Q.; validation, D.S., Z.W. and X.Q.; formal analysis, D.S.; investigation, J.G., Q.C., Z.W. and K.L.; resources, D.S.; data curation, D.S. and K.L.; writing—original draft preparation, X.Q. and J.G.; writing—review and editing, D.S., K.L., Z.W. and X.Q.; visualization, K.L., Q.C. and D.S.; supervision, D.S. and J.G. All authors have read and agreed to the published version of the manuscript.

**Funding:** This research was funded by the Central Government to Guide Local Science and Technology Development fund Projects of Hubei Province (2022BGE267), the Basic Research Type of Science and Technology Planning Projects of Xiangyang City (2022ABH006759), the Hubei Superior and Distinctive Discipline Group of "New Energy Vehicle and Smart Transportation" (XKTD072023), and the Central Government of Hubei Province to Guide Local Science and Technology Development (2020ZYYD001).

**Institutional Review Board Statement:** Not applicable.

**Informed Consent Statement:** Not applicable.

**Data Availability Statement:** The data presented in this study are available on per request to the corresponding author.

**Conflicts of Interest:** The authors declare no conflict of interest.

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
