# Peer review of "Research on an Improved Rule-Based Energy Management Strategy Enlightened by the DP Optimization Results"

_sustainability, doi:10.3390/su151310472_

Round 1
Reviewer 1 Report
This paper reported proposed an improved rule-based (IRB) strategy enlightened by DP strategy, which can optimize the PHEV. The results show that the fuel consumption of IRB strategy under CLTC driving conditions is effectively decreased. Thus, this manuscript can be considered for publication after minor revision.
1. Make sure all abbreviations are written out in full the first time used, such as SOC, CLTC and so on. This is particularly important in the abstract and in the conclusions, but work through the entire ms carefully from this perspective.
2. The energy management strategy based on different systems should be elaborated. Some references may be cited to enrich this part, such as Nano Research Energy, 2022, 1: e9120014; https://doi.org/10.26599/NRE.2023.9120062.
3. Figure 9 should be drawn by professional software.
4. Please list the parameters of how to calculate the engine energy loss.
5. Please provide the some future perspectives and outlooks in conclusion.
The language should be checked carefully.
Author Response
Comments:
This paper reported proposed an improved rule-based (IRB) strategy enlightened by DP strategy, which can optimize the PHEV. The results show that the fuel consumption of IRB strategy under CLTC driving conditions is effectively decreased. Thus, this manuscript can be considered for publication after minor revision.
Response: We appreciate your suggestion. The relevant modifications have been made in the new manuscript.
- Make sure all abbreviations are written out in full the first time used, such as SOC, CLTC and so on. This is particularly important in the abstract and in the conclusions, but work through the entire ms carefully from this perspective.
Response: We agree with your comments. It has been revised in the new manuscript.
- The energy management strategy based on different systems should be elaborated. Some references may be cited to enrich this part, such as Nano Research Energy, 2022, 1: e9120014; https://doi.org/10.26599/NRE.2023.9120062.
Response: We appreciate your suggestion. However, this literature may not be relevant to our study.
- Figure 9 should be drawn by professional software.
Response: We appreciate your suggestion. The relevant modifications have been made in the new manuscript.
4、Please list the parameters of how to calculate the engine energy loss.
Response: The relevant modifications have been made in the new manuscript.
- Please provide the some future perspectives and outlooks in conclusion.
Response: We appreciate your suggestion. The relevant modifications have been made in the new manuscript.
Reviewer 2 Report
The work is good, some modification is required in the abstract and conclusion sections. Specify the novelty of the work in the abstract and highlight the outcome of the paper in the conclusion section.
Figure 3 caption is not mentioned on page 4.
Add some latest references in the reference section.
Some of the sentences can be improved.
Author Response
Comments:
The work is good, some modification is required in the abstract and conclusion sections. Specify the novelty of the work in the abstract and highlight the outcome of the paper in the conclusion section.
Response: We appreciate your suggestion. The relevant modifications have been made in the new manuscript.
1、Figure 3 caption is not mentioned on page 4.
Response: The relevant modifications have been made in the new manuscript.
2、Add some latest references in the reference section.
Response: The relevant modifications have been made in the new manuscript.
Reviewer 3 Report
Following are my comments:
Introduction: It may be very useful to have advantages and disadvantages of RB and DP strategies laid out more clearly. For example, RB is not fuel efficient but requires lesser computing power and DP is fuel efficient but computation might be more complex. Hence, an intermediary strategy (IRB) is required to combine advantages from both. This will help emphasize on the story about why IRB is required.
Please provide full form of abbreviations before first use. (Line 102: BFSC, Line 170: CLTC)
Section 2.3:
What are units for the terms used? SOC is Wh? Qb is Ah? please specify.
What is the take-away from this section? How does the co-relation between battery SOC and internal resistance apply to the model?
Section 2 needs to address how the different approaches apply the model to the vehicle. For example, in vehicle, engine fuel consumption, battery, and ISG models what are the input parameters and output parameters and how they apply to vehicle function? A block diagram or flow chart would be helpful.
Line 139: What is the negative effect of increasing internal resistance of the battery?
Section 3.2: It is difficult to track the flow for this section. Please elaborate more for better understanding of the reader.
Line 240: What is the justification for 2%?
Line 278: Typo: RB instead of IRB
Section 4: It'll be useful to have a flowchart for the IRB strategy.
There are many typos that must be addressed. Grammar and flow of sentences can be improved.
Author Response
Comments:
Introduction: It may be very useful to have advantages and disadvantages of RB and DP strategies laid out more clearly. For example, RB is not fuel efficient but requires lesser computing power and DP is fuel efficient but computation might be more complex. Hence, an intermediary strategy is required to combine advantages from both. This will help emphasize on the story about why IRB is required.
Response: We appreciate your suggestion. The relevant modifications have been made in the new manuscript.
1、Please provide full form of abbreviations before first use. (Line 102: BFSC, Line 170: CLTC)
Section 2.3:
Response: The relevant modifications have been made in the new manuscript.
2、What are units for the terms used? SOC is Wh? Qb is Ah? please specify.
Response: The relevant modifications have been made in the new manuscript.
3、What is the take-away from this section? How does the co-relation between battery SOC and internal resistance apply to the model? Section 2 needs to address how the different approaches apply the model to the vehicle. For example, in vehicle, engine fuel consumption, battery, and ISG models what are the input parameters and output parameters and how they apply to vehicle function? A block diagram or flow chart would be helpful.
Response: We appreciate your suggestion. The relationship between the input and output of the mathematical model of key components can be obtained through the connection mode of key components in Figure 1, which is a common modeling method for studying hybrid electric vehicles.
4、Line 139: What is the negative effect of increasing internal resistance of the battery?
Response: According to Joule's law, under the same battery current, the greater the internal resistance, the more heat it generates, that is, the greater the energy losses of the battery.
5、Section 3.2: It is difficult to track the flow for this section. Please elaborate more for better understanding of the reader.
Response: We appreciate your suggestion. The relevant modifications have been made in the new manuscript.
6、Line 240: What is the justification for 2%?
Response: It depends on your research vehicle’s battery size; 3% may also be good for our research.
7、Line 278: Typo: RB instead of IRB
Response: The relevant modifications have been made in the new manuscript.
8、Section 4: It'll be useful to have a flowchart for the IRB strategy.
Response: We appreciate your suggestion. The relevant modifications have been made in the new manuscript.
Round 2
Reviewer 3 Report
I appreciate the authors in addressing most of my comments.
4. The internal resistance's relation to the increased energy loss due to heat dissipation can be included as few sentences in the article to emphasize the negative impact of increasing internal resistance with decreasing SOC. If the graph in Figure 3 can be flipped so that SOC decreases from 1 to 0 and how that affects internal resistance, it would be more analytically representative.
Author Response
Comments:
I appreciate the authors in addressing most of my comments.
4、The internal resistance's relation to the increased energy loss due to heat dissipation can be included as few sentences in the article to emphasize the negative impact of increasing internal resistance with decreasing SOC. If the graph in Figure 3 can be flipped so that SOC decreases from 1 to 0 and how that affects internal resistance, it would be more analytically representative.
Response: We appreciate your suggestion. The relevant modifications have been made in the new manuscript.
